# Cost-effective Data Labelling for Graph Neural Networks

## ABSTRACT

Active learning (AL), that aims to label limited data samples to effectively train the model, stands as a very cost-effective data labelling strategy in machine learning. Given the state-of-the-art performance GNNs have achieved in graph-based tasks, it is critical to design proper AL methods for graph neural networks (GNNs). However, existing GNN-based AL methods require considerable supervised information to guide the AL process, such as the GNN model to use, and initially labelled nodes and labels of newly selected nodes. Such dependency on supervised information limits both flexibility and scalabilty. In this paper, we propose an *unsupervised, scalable and flexible* AL method – it incurs low memory footprints and time cost, is flexible to the choice of underlying GNNs, and operates without requiring GNN-model-specific knowledge or labels of selected nodes. Specifically, we leverage the commonality of existing GNNs to reformulate the unsupervised AL problem as the Aggregation Involvement Maximization (AIM) problem. The objective of AIM is to maximize the involvement or participation of all nodes during the feature aggregation process of GNNs for nodes to be labelled. In this way, the aggregated features of labelled nodes can be diversified to a large extent, thereby benefiting the training of feature transformation matrices which are major trainable components in GNNs. We prove that the AIM problem is NP-hard and propose an efficient solution with theoretical guarantees. Extensive experiments on public datasets demonstrate the effectiveness, scalability and flexibility of our method. Our study is highly relevant to the track "Graph Algorithms and Modeling for the Web" since we focus one of the major listed topics "Graph Embedding and GNNs for the Web" and AL for GNNs, as an important research problem, is faced by aforementioned challenges to be tackled in this paper.

**ACM Reference Format:**
Anonymous Author(s). 2023. Cost-effective Data Labelling for Graph Neural Networks. In *Proceedings of ACM Conference (Conference'17).* ACM, New York, NY, USA, 12 pages. https://doi.org/10.1145/nnnnnnn.nnnnnnn

## 1 INTRODUCTION

Graph embedding, which aims to learn low-dimensional representations for nodes in graphs, has been a popular tool for solving various graph-based tasks (e.g., community search [15], entity alignment [27], node classification [18], and time series forecasting [11]) in recent years. Among all embedding techniques, Graph Neural Networks (GNNs) have achieved state-of-the-art performance [7, 9, 11, 15, 20, 38]. However, to train powerful GNNs, a large

number of labeled nodes are needed and it requires costly manual labeling from experts. Thus, Active Learning (AL) [2, 4, 12, 17, 29, 35], which aims to label limited training data samples so as to maximize the model performance, has received considerable attention.

Despite significant progress being made, there remains a lack of practical and efficient GNN-based AL methods for three reasons. First, traditional AL methods [5, 31, 37, 42, 50] mainly consider learning models on independent and identically distributed (i.i.d) data (e.g., images, text or tabular data [46]). However, graphs, which are built upon nodes' interactions, are not i.i.d and connected nodes tend to have similar labels. Thus, traditional successful AL techniques in other fields cannot be trivially extended to benefit GNN training. Second, many GNN-based AL methods [6, 16, 21, 45] are supervised. That means in each iteration, they need to not only select but also label some nodes with a GNN, and then use the newly labelled information to update/train the GNN to guide subsequent selection. Moreover, they may require a 'warm up' phase in which some nodes are labelled and assumed to be available prior to the AL process and follow some distributions (e.g., balanced labels). This supervised setting and 'warm up' procedure are not practical: (1) it implies prior knowledge of the GNN to use (GNN-model-specific node selection), which does not account for scenarios where the GNN is unknown during the AL process or labelled nodes may be repetitively used by different GNNs later for various downstream tasks; (2) it might be ineffective to rely on the GNN, which is initially inaccurate and needs to be trained will sufficient labelled information, to guide node selection. Such a strategy can easily make wrong decisions at early stages and trigger a domino effect that impacts subsequent selection; (3) obtaining initial labeled nodes to 'warm up' models can be challenging in various domains, such as medical research (due to privacy concern) [3] and autonomous driving (due to the exorbitantly expensive labelling process) [10]. Third, existing methods suffer from scalability issues (e.g., high running time and/or memory cost), since AL needs to either train a model [6, 16, 45] or perform expensive matrix operations [41, 48]. **Our objective** is to propose an *unsupervised, scalable, and flexible* GNN-based AL method. By 'scalable', we aim to propose an AL algorithm with low memory footprints and time cost. By 'unsupervised', it does not require initially labelled nodes to 'warm up', prior knowledge of a specific GNN for downstream usage nor labels of newly selected nodes to guide future selection. Instead, its design only relies on the general knowledge of GNNs (i.e., the high-level process of the multi-layer message passing). By 'flexible', its design can greatly benefit the later training (after AL) of many GNNs that will be used for downstream tasks. A comparison of the supervised strategy and our unsupervised strategy is shown in Fig. 1.

To achieve the above, we design our methods by leveraging the commonality of existing GNNs for node selection. Specifically, most GNNs learn transformation matrices that linearly transform aggregated features from neighborhood. These matrices constitute the main trainable parameters and notably impact model performance.

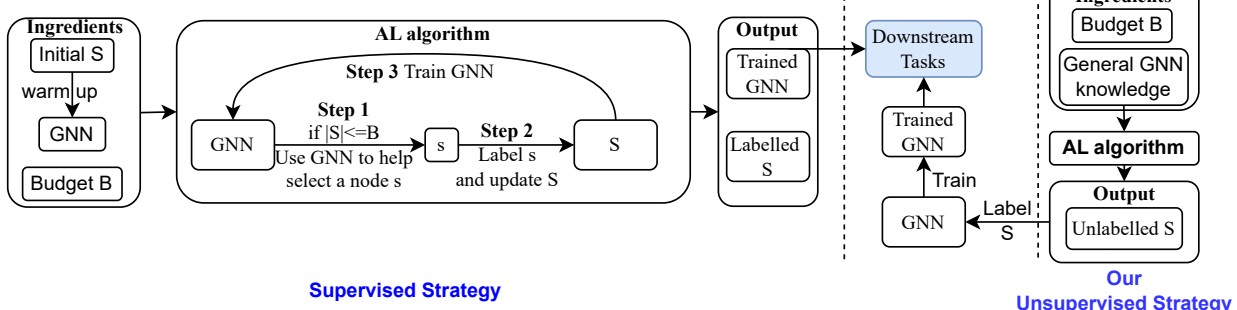

**Figure 1: The supervised strategy (by the GNN model, labels of iteratively selected nodes and a potential initially-labelled set) v.s. our unsupervised strategy which does not require the GNN model nor node labels during the AL process**

Hence, our emphasize selecting nodes whose label information can offer the maximum benefit to the training of these matrices.

We argue that training powerful transformation matrices needs diversified aggregated features and the diversification is positively correlated with nodes' involvement in the aggregation process. Therefore, we cast the AL problem into a new problem called Aggregation Involvement Maximization (AIM), which aims to select a limited number of nodes such that the feature aggregation process for these selected nodes can lead to the maximum involvement of *all* nodes in the graph. Accordingly, we propose efficient and effective algorithms with theoretical guarantees to solve this problem. Our contributions are summarized below:

- We study the *unsupervised active learning* research problem for graph data. To derive an effective solution for it, we reformulate it as the Aggregation Involvement Maximization (AIM) problem. In the new problem, we propose a novel definition of aggregation involvement to encourage diversification of aggregated features of selected nodes and thus benefit the training of the feature transformation matrices.
- We prove that the AIM problem is NP-hard and the objective is monotone and submodular, and then propose a greedy solution which adopts an early termination technique to efficiently produce solutions with an approximation ratio of $1 - 1/e$.
- We conduct extensive experiments on real-world datasets to show that our method outperforms state-of-the-arts in terms of effectiveness (up to 19.4% higher accuracy) while maintaining high efficiency (up to five-orders-of-magnitude speedups) and low memory footprints (save up to 102.4x memory space), and is flexible to the choice of downstream GNNs.

## 2 RELATED WORK

**Common AL techniques**. AL provide solutions for selecting the most valuable samples for labeling so as to optimize the model performance. Uncertainty sampling is considered in [42, 50] to select nodes and it has the most uncertain model prediction. Some works [5, 31] select samples based on the extent to which the training models disagree. Furthermore, there also exist density-based [37], clustering-based [13, 33] and diversity-based solutions [23, 40] which select samples based on different criteria. Traditional AL

strategies mainly focus on learning models on independent and identically distributed (i.i.d) data, and thus fail to consider the graph topology information.

**GNN-based AL techniques**. Different strategies of GNN-based AL have been proposed to incorporate topology information in graph-structured data. AGE [6] and ANRMAB [16] take into account uncertainty, density and node degree to select nodes for labeling. ANRMAB improves AGE by a multi-armed bandit mechanism for enhanced decision making upon node selection. FeatProp [41] generates output node features using a simplified GCN and then extends K-Means to select nodes in the cluster centers for labeling. GPA [21] involves joint training on several source graphs and employs reinforcement learning to learn a transferable active learning policy. ALG [45] decouples the GNN model for the efficiency reason and considers maximizing the effective reception field. Grain [48] further generalizes the reception field by considering diversified feature influence. There are some works [46, 47] on variants of AL (e.g., noisy labelling oracles) which are orthogonal to our study.

Most existing studies on the classical AL problem are either supervised or semi-supervised. Specifically, methods like [6, 16, 21, 45] are supervised since the AL process is supervised by the GNN model and newly selected and labelled nodes. We consider methods like [21, 48] semi-supervised – while they do not need to explicitly train the GNN, they still rely on node features generated by the GNN-model-specific aggregation matrix. Moreover, their node feature generation and node selection process can be very expensive (e.g., large-scale matrix multiplication and Jacobian matrix estimation [48]). In contrast, our unsupervised AL algorithm incurs low memory footprints and time cost, and it does not require GNN-model-specific knowledge.

## 3 PRELIMINARY AND PROBLEM DEFINITION

In this section, we will describe important notations, GNNs and the problem formulation. We denote a graph as $G = (V, E)$ where $V$ and $E$ refer to the node set and edge set respectively. We set $N = |V|$ and denote the adjacency matrix of the graph as $A \in \mathbb{R}^{N \times N}$. Each node $v$ has an input embedding vector $x \in \mathbb{R}^d$ and all node vectors together form the embedding matrix $X \in \mathbb{R}^{N \times d}$. Each node $v$ is also

associated with a one-hot vector $y \in \mathbb{R}^C$, where $C$ is the number of classes and the $c^{th}$ element is 1 only if $v$ belongs to class $c$.

**Graph Neural Networks (GNNs).** GNNs [19, 25, 38] define a multi-layer message passing process. In this process, the feature representation of a *target node* in the next layer is the aggregation result of the current-layer features of nodes in the neighborhood. Below is a general recursive function for message passing:

$$X^{(K)} = f(X^{(K-1)}, T, \Theta^{(K)}, X), \qquad (1)$$

Here, $X$ is the initial embedding matrix, $X^{(K)}$ is the output embedding matrix, $\Theta^{(K)}$ is the *feature transformation matrix at* layer $K$, and $T$ is the *feature aggregation matrix*.

**Roles of feature transformation and aggregation matrices.** In GNNs, the main trainable parameters are feature transformation matrices at layers that linearly transform aggregated representations. The feature aggregation matrix $T$ specifies how features of nodes in neighborhood are aggregated to the target node.

Existing GNNs share the above procedure but primarily vary in how they define and leverage the feature aggregation matrix. E.g., Graph Convolution Network (GCN) [25] defines $f$ as:

$$X^{(K)} = \delta(T X^{(K-1)} \Theta^{(K)}),$$

where $T = \tilde{D}^{\frac{1}{2}} \tilde{A} \tilde{D}^{\frac{1}{2}}$, $\tilde{A} = A + I_N$, $I_N$ is the identity matrix, $\tilde{D}$ is the diagonal degree matrix of $\tilde{A}$ and $\delta(\cdot)$ is the activation function.

After this recursive process, a prediction function (e.g., softmax) is applied to the output embedding matrix $X^{(K)}$ for the downstream task (e.g., node classification). By considering the task, graph topology and features jointly, existing studies define $T$ differently (e.g., symmetric transition matrix [25], the random walk transition matrix [26, 39], triangle-based adjacency matrix [14], and powers of the adjacency matrix [8, 49]).

We assume that the entire node set $V$ is divided into three sets, namely the training set $V_{train}$, the validation set $V_{val}$ and the test set $V_{test}$, and the training algorithm $M$ is a GNN. The problem we study in this paper is defined as below.

**Definition 1 (Active Learning).** *Given a loss function $\ell$ and a node set $V_{train}$ which does not have labelled nodes, the aim of active learning is to select an optimal seed set $S^*$ of $B$ seed nodes from $V_{train}$ to label, such that the lowest loss on the test set $V_{test}$ can be achieved by training the algorithm $M$ with the labeled seed set $S^*$ only. That is,*

$$S^* = \underset{S \in V_{train} \wedge |S| = B}{\arg\min} \; \mathbb{E}_{v_i \in V_{test}} [\ell(y_i, P(\hat{y}_i | x_i, M_S))],$$

*where $M$ is not known during the node selection process, $M_S$ is the algorithm $M$ trained under the supervision of $S$ and $P(\hat{y}_i | x_i, M_S))$ is the label distribution predicted by $M_S$.*

## 4 AGGREGATION INVOLVEMENT MAXIMIZATION

As explained in Section 3, after the recursive function (Eq 1) is constructed, the performance of a GNN mainly depends on how well the feature transformation matrices are trained. To train powerful feature transformation matrices, we cast the AL problem to the Aggregation Involvement Maximization (AIM) problem, and accordingly, we propose highly effective and efficient solutions to solve the AIM problem. We will first describe the rationale behind

this cast and how the AIM problem is formulated (Sec. 4.1), followed by a theoretical proof of the NP-hardness and properties of the AIM problem (Sec. 4.4), and ultimately, our solution (Sec. 4.5).

### 4.1 From AL to Aggregation Involvement Maximization (AIM)

**Diverse aggregated features: key to high-quality training.** Since the primary trainable parameters in GNNs are the transformation matrices that operate on aggregated features, it is essential to select seed nodes whose aggregated features can benefit the training of these matrices to the maximum extent. Thus, given a limited budget to select seeds, an effective strategy is to promote diversity in the distribution of aggregated features among the selected seeds. These diverse aggregated features cover representative aggregation cases in terms of graph topology and the input features of nodes, helping ensure the generalization power of the learned parameters.

**Diverse aggregated features to aggregation neighborhoods.** While GNNs vary in how they aggregate features from neighborhoods, they all require training transformation matrices whose quality depends on the diversity of the seeds' aggregated features. To ensure compatibility with various GNNs, even when the specific GNN is unknown during seed selection, we must diversify aggregation neighborhoods. This diversity implicitly maintains the diversification of aggregated features. For instance, suppose we consider aggregation from one-hop neighbors and choosing two seeds in Figure 2. If choosing $f$ and $c$ as two seeds, their aggregated features will be similar given their similar aggregation neighborhoods. Choosing $c$ and choosing $g$ can be a better solution since the aggregation neighborhoods are more diverse. As a result, the aggregated features are also diverse as well no matter which GNN is used (e.g., for seed $c$, one GNN may focus on aggregating from node $f$ and $j$ whereas another GNN may focus on node $m$ and $b$).

**Diversifying neighborhood via involvement maximization.** We quantify the neighborhood diversification based on node *involvement*, which indicates a node's contribution to generating aggregated features for seeds. Nodes that appear within $k$-hop naturally participate in the aggregation process, and hence are *directly* involved. Due to graph topology's homophily effect, nodes 'close' to each other tend to have similar representations. To promote neighborhood diversification, we encourage nodes outside the located aggregation neighborhoods but close in graph topology to be *indirectly* involved. Maximizing nodes' involvement naturally 'spreads out' aggregation neighborhoods, enhancing diversification.

To facilitate the illustration, we define the problem first before introducing each notation involved. Let $S$ denote a seed set, $\delta_k(S)$ denote the set of directly involved nodes, $|\delta_k(S)|$ denote the total direct involvement score (see Section 4.2), $I_v(SIM_v(S))$ denote the indirect involvement score of $v \in V \setminus \delta_k(S)$ (see Section 4.3), we define the total involvement and the problem as below:

**Definition 2 (Total Aggregation Involvement).** *The total aggregation involvement $Q(S)$ achieved by selecting $S$ as the seed set is the sum of the total direct and indirect involvement scores. Namely, $Q(S) = |\delta_k(S)| + \sum_{v \in V \setminus \delta(S)} I_v(SIM_v(S))$.*

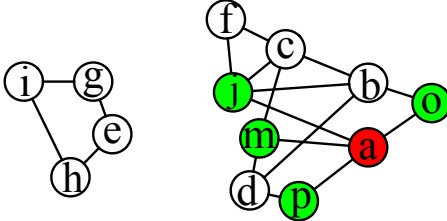

**Figure 2: A graph where $K = 1$, node $a$ is the current chosen seed and green nodes are the aggregation sources. Nodes within 1-hop away from nodes in the aggregation neighborhoods of seeds have indirect involvement.**

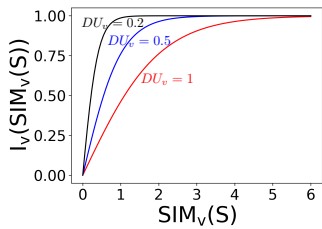

**Figure 3: Curves of the function $I_v(\cdot)$ with different $DU_v$**

DEFINITION 3 (AGGREGATION INVOLVEMENT MAXIMIZATION (AIM)). *Given a fixed labeling budget $B$, we aim to find a set $S^*$ of $B$ nodes such that the total aggregation involvement $Q(S^*)$ is the maximum. That is, $S^* = \arg\max_{S \subseteq V_{train}, |S| \leq B} Q(S)$.*

### 4.2 Direct Aggregation Involvement

We define the aggregation neighborhood of a seed $s$ as the set $\delta_k(s)$ of nodes which are within $k$-hop away from $s$. Considering that we have no knowledge of the feature aggregation matrix of the downstream GNN, we assume that a node $u$ has direct (aggregation) involvement if it appears in $\delta_k(s)$ of a seed $s$, since the node $u$ may directly propagate its feature to $s$. Thus, all nodes with direct involvement can be represented as:

$$\delta_k(S) = \cup_{s \in S} \delta_k(s). \tag{2}$$

Since many GNNs achieve high performance by setting the number of layers $k = 2$ [19, 20, 24, 25, 43], it is sufficient to control $k \leq 2$. Even when the underlying GNN needs to reach nodes more than 2 hops away (e.g., $k = 10$ in APPNP [26]), controlling the aggregation neighborhood within two hops is still highly effective as demonstrated in the experiments.

### 4.3 Indirect Aggregation Involvement

We define the indirect aggregation involvement based on involvement-based similarity. In what follows, we will give a high-level overview of the connection between these two and then describe how to quantify the involvement-based similarity (Section 4.3.1) and use this similarity to compute the indirect involvement (Section 4.3.2).

**Indirect involvement and involvement-based similarity**. Since the labeling budget $B$ is limited, there could exist considerable nodes that are outside of $\delta_k(S)$. These nodes could also be involved to estimate the diversification of the aggregation neighborhoods of seeds and such involvement is indirect. The indirect involvement

of a node $v$ is positively related to the relationship between $v$ and the nodes in $\delta_k(S)$, and we call this relationship as the *involvement-based similarity* between $v$ and $\delta_k(S)$. If there are many nodes in $\delta_k(S)$ and close to $v$ in terms of both the graph topology and embedding space, this similarity should also be high.

The intuition of indirect involvement is that, when the involvement-based similarity of $v$ is high and we are selecting a new seed $s$ into $S$, we should pay less attention to the candidates whose aggregation neighborhoods include $v$. This is because nodes, close to $v$ or appearing in $\delta_k(S)$, may be very likely to appear in candidates' aggregation neighborhoods. These neighborhoods may heavily overlap with or very close to the ones of chosen seeds and cannot help further increase aggregated features' diversification.

EXAMPLE 1. *In Fig. 2, node $a$ has been chosen as a seed and choosing $b$ as a new seed will not really benefit the training of the transformation matrix, since many of its aggregation sources (e.g., node $c$ and $d$) already have very high indirect involvement and its aggregation neighborhood notably overlaps with the one of node $a$. On the other hand, node $e$ and the nodes in its neighborhood have very little involvement. Thus, choosing $e$ instead of $b$ as a new seed can better increase total nodes' involvement and help diversify the distributions of selected aggregation neighborhoods.*

#### 4.3.1 Involvement-based Similarity.
Due to the homophily effect, we should mainly focus on the nodes in near neighborhood of $v$ to compute the involvement-based similarity. Specifically, let $\mathcal{H}(v)$ (whose size will be discussed shortly) denote the set of nodes within certain hops away from of $v$, to compute the involvement-based similarity between $v$ and $\delta_k(S)$, we should focus on the nodes in $\delta_k(S) \cap \mathcal{H}(v)$. We call such nodes in $\delta_k(S) \cap \mathcal{H}(v)$ as *involvement-based relevant nodes* of $v$ and each pair between $v$ and one of such nodes as an *involvement-based relevant pair*. When the context is clear, we use 'involvement-based relevant' and 'relevant' interchangeably. We define the involvement-based similarity $SIM_v(S)$ between $v$ and $\delta_k(S)$ as the sum of similarities of all relevant pairs:

$$SIM_v(S) = \sum_{r \in \delta_k(S) \cap \mathcal{H}(v)} sim_v(r) \tag{3}$$

where $sim_v(r)$ denotes the similarity of embeddings between $v$ and the relevant node $r$ and can be computed with different similarity metrics (e.g., cosine similarity and distance-based similarity) to cater for datasets with different characteristics. If it is distance-based (e.g., Euclidean distance based) similarity, $sim_v(r)$ can be computed as $\frac{1}{1+Dis(v,r)}$ where $Dis(v, r)$ is the normalized distance.

EXAMPLE 2. *In Figure 2, if $S = a$ and $\delta_k(a)$ only includes nodes 1 hop away from $a$, $\delta_k(S)$ consists of all green nodes (i.e., nodes $j, m, o$ and $p$), $\mathcal{H}(c) = \{b, f, j, m\}$, the set of relevant nodes of $c$ is $\mathcal{H}(c) \cap \delta_k(S) = \{j, m\}$, and the pair $(c, j)$ is a relevant pair.*

**Size of $\mathcal{H}(v)$.** To ensure the scalability and effectiveness, we need to carefully decide the maximum hop between any node in $\mathcal{H}(v)$ and $v$. We conduct two studies: (1) how are similarities between nodes in the same aggregation neighborhood distributed? (2) how are the sizes of neighborhoods within different hops distributed? For study 1, we randomly sample 1000 nodes from the dataset, enumerate all pairs of nodes within 2-hop away from the same sampled nodes and summarize their cosine similarities based on

their graph distance. For study 2, we summarize the average size of nodes which are within certain hops away from these sampled nodes. Table 1 and Table 2 show the results for study 1 and study 2 on all datasets in our experiments respectively. It is obvious that similar node pairs tend to be directly connected in most cases and the size of $\mathcal{H}(\cdot)$ increases significantly as the maximum hop grows. Therefore, it is sufficient to just include 1-hop neighbors in $\mathcal{H}(v)$.

**Table 1: The average cosine similarity between nodes in the same aggregation neighborhood with different distances**

| Dataset | 1-hop | 2-hop | 3-hop | 4-hop |
|---------|-------|-------|-------|-------|
| Cora | 0.150 | 0.095 | 0.082 | 0.069 |
| Citeseer | 0.162 | 0.095 | 0.079 | 0.067 |
| Pubmed | 0.280 | 0.193 | 0.159 | 0.117 |
| Co-Phy | 0.473 | 0.270 | 0.188 | 0.113 |
| Flickr | 0.321 | 0.310 | 0.312 | 0.314 |
| Arxiv | 0.727 | 0.708 | 0.710 | 0.714 |

**Table 2: The average percentage of the # of the nodes, within certain hops away from sampled nodes, over the graph size**

| Dataset | 1-hop | 2-hop | 3-hop | 4-hop |
|---------|-------|-------|-------|-------|
| Cora | 0.14 | 1.31 | 22.50 | 44.90 |
| Citeseer | 0.08 | 0.46 | 8.87 | 21.05 |
| Pubmed | 0.02 | 0.30 | 8.29 | 51.92 |
| Co-Phy | 0.04 | 0.67 | 27.64 | 59.63 |
| Flickr | 0.01 | 0.74 | 13.05 | 65.44 |
| Arxiv | 0.01 | 2.90 | 35.14 | 47.28 |

*4.3.2 Definition of Indirect Involvement.* We define the indirect involvement as a function of the involvement-based similarity defined above. Specifically, we incrementally update the definition of indirect involvement by considering neighborhood diversification, node-dependent similarity significance and similarity normalization. In what follows, we will introduce these considerations and the corresponding definition development of indirect involvement.

**Consideration 1 - Neighborhood Diversification**. To ensure that the chosen seeds and their aggregation neighborhoods are diversified in the embedding space, we should encourage the solution to choose seeds whose aggregation neighborhoods are close to nodes with relatively low indirect involvement. To achieve this purpose, given the same increment of the involvement-based similarity score, a node whose current indirect involvement is smaller needs to achieve a larger increment of the indirect involvement score. Thus, the indirect involvement of $v$ is defined as below:

$$I_v(SIM_v(S)) = \frac{2}{1 + e^{-SIM_v(S)}} - 1 \qquad (4)$$

The indirect involvement $I_v(\cdot)$ of $v$ is positively related to the involvement-based similarity $SIM_v(S)$ and its value is controlled between 0 and 1. The exponential function encourages the solution to consider nodes with little indirect involvement.

EXAMPLE 3. *In Figure 2, suppose we consider choosing a new seed between b and e, $S = \{a\}$, $SIM_f(S) = 0.5$, $I_f(SIM_f(S)) = 0.24$, $SIM_i(S) = 0$ and $I_i(SIM_i(S)) = 0$. We should encourage the solution to choose e to diversify the aggregation neighborhood. Since*

*choosing e and b respectively can both directly involve two new nodes (i.e., choosing e directly involves g and h, and choosing b directly involves c and d), we need to ensure that choosing e brings more indirect involvement increment. Equation 4 helps to achieve this purpose. For instance, if choosing e and b gives the same increment (say 0.5) to the involvement-based similarity score of i and f respectively, choosing e achieves a larger indirect involvement increment since $I_i(SIM_i(S \cup \{e\})) - I_i(SIM_i(S)) = I_i(0.5 + 0) - I_i(0) = 0.24 \geq I_f(SIM_f(S \cup \{b\})) - I_f(SIM_f(S)) = I_f(0.5 + 0.5) - I_f(0.5) = 0.22$.*

**Consideration 2 - Similarity Significance**. Consideration 1 helps diversify the selected seeds when nodes' current indirect involvement scores are very different. However, when the involvement scores are very similar, we need to carefully evaluate the impact of the involvement-based similarity score on the indirect involvement score of each node $v$ in a finer granularity, by looking into the constituent of the involvement-based similarity - the similarities of relevant pairs. Specifically, the significance of a pairwise similarity score should be node-dependent and the same involvement-based similarity score increment may have different impacts on nodes' indirect involvement scores even when the current indirect involvement scores of these nodes are the same.

EXAMPLE 4. *Suppose the pairwise similarities between u and each node in $\mathcal{H}(u)$ are 0.5, 0.01, 0.01 and 0.01 respectively, and the ones between v and each node in $\mathcal{H}(v)$ are 0.5, 0.5, 0.5 and 0.5 respectively. Apparently, the same pairwise similarity of 0.5 should have different impacts on the indirect involvement score of u and v. The 0.5 increment of the involvement-based similarity score should result in much more the indirect involvement score increment of u than it does for node v.*

Therefore, to capture the significance of pairwise similarities, we should consider their distributions instead of their absolute values, and it can be captured by Distribution Uniformity ($DU$) [1]. $DU$ originates from irrigation and is a measure of how uniformly water is applied to the area being watered. The most common measure of $DU$ is the average of the lowest quarter of samples divided by the average of all samples. The higher $DU$ is, the more uniform the coverage of the measured area is. If all samples are equal, $DU$ is 1.0. In our problem, for a node $v$, we treat $sim_v(u)$ between $v$ and each node $u$ in $\mathcal{H}(v)$ as a sample to compute $DU_v = \frac{\text{the average of the lowest quarter in } \{sim_v(u)|u \in \mathcal{H}(v)\}}{\text{the average of all samples in } \{sim_v(u)|u \in \mathcal{H}(v)\}}$.

When the size $\mathcal{H}(v)$ is small (e.g., 2), we use the first half instead of the lowest quarter for the numerator. Then, the indirect involvement can be defined as below:

$$I_v(SIM_v(S)) = \frac{2}{1 + e^{-\frac{SIM_v(S)}{DU_v}}} - 1, \qquad (5)$$

EXAMPLE 5. *Figure 3 shows the curves of the function $I_v(\cdot)$ with different $DU_v$. A smaller $DU_v$ indicates that some pairwise similarities 'stand out' in $\{sim_v(u)|u \in \mathcal{H}(v)\}$ and achieve larger significance. The design of the function (i.e., Equation 5) gives a steeper slope with a smaller $DU_v$ which can encourage the solution to realize the similarity significance and hence give more consideration to outstanding pairwise similarities during seed selection.*

**Consideration 3 - Similarity Normalization**. Previous formulations of the indirect involvement function do no consider the maximum possible involvement-based similarity that a node can

achieve. If the indirect involvement score corresponding to the maximum possible involvement-based similarity is very close to 1, considerable similarity increment may barely increase the indirect involvement score since the function gradient is too small.

EXAMPLE 6. *Suppose* $\{sim_v(u)|u \in \mathcal{H}(v)\}$ *consists of 20 pairwise similarities of the same score 0.9. In this case,* $DU_v = 1$ *and the corresponding function* $I_v(\cdot)$ *will become the red curve in Figure 3. Since the maximum* $SIM_v(S)$ *is* $20 \times 0.9 = 18$, *the value* $I_v(\cdot)$ *is very close to 1 (i.e.,* $I_v(\cdot) = 0.9999$*) if there are 12 relevant pairs (i.e.,* $SIM_v(S) = 10.8$*). Therefore, after* $SIM_v(S) = 10.8$, *there could be a large amount of increment of the* $SIM_v(S)$ *during the seed selection but the corresponding increment of* $I_v(\cdot)$ *is negligible. Thus, the solution may fail to distinguish the quality of each node as a potential seed.*

Thus, we need to normalize these similarities such that their total sum is within a reasonable range $[0, \eta]$ and thus each unit of similarity increment would bring non-negligible indirect involvement score increment. To obtain $\eta$, we first set the threshold $t$ (e.g., 0.9999) of the indirect involvement score that a node can achieve. Correspondingly, we can have $\eta$ which satisfies $2/(1 + e^{-\frac{\eta}{DU_v}}) - 1 = t$. After obtaining $\eta$, we can adjust Equation 5 by including the normalization factor $NF_v = \frac{\eta}{\sum_{u \in \mathcal{H}(v)} sim_v(u)}$:

$$I_v(SIM_v(S)) = \frac{2}{1 + e^{-\frac{SIM_v(S)}{DU_v} \times NF_v}} - 1. \quad (6)$$

Via this way, $SIM_v(S) \times NF_v$ and $I_v(SIM_v(S))$ will be controlled within $\eta$ and $t$ respectively.

## 4.4 Theoretical Analysis

In this section, we will prove that the AIM is NP-hard and the objective is monotone and submodular (the proof is in Appendix A).

THEOREM 1. *The AIM is NP-hard.*

PROOF. We prove the NP-hardness via a reduction from the NP-hard Maximum Coverage problem.

DEFINITION 4 (MAXIMUM COVERAGE). *Given a set* $U = \{u_1, u_2, ..., u_{|U|}\}$ *of elements, a collection* $C$ *of subsets* $L_1, L_2, ..., L_{|C|}$ *of* $U$, *and an integer* $k$, *the goal is to find an optimal collection* $O^* \subseteq C$ *such that* $|O^*| \leq k$ *and* $|\cup_{L \in O^*} L|$ *is maximized.*

Given an arbitrary instance of the maximum coverage problem, we define a corresponding directed bipartite graph with $|C| + |U|$ nodes. Specifically, a node $i$ corresponds to a set $L_i \in C$ and a node $j$ corresponds to $u_j \in U$. Whenever $u_j \in L_i$, we create an edge $(i, j)$. Next, we set $k = 1$ and the target similarity $t = 0$ such that maximizing $Q(\cdot)$ is equivalent to maximizing $|\delta_k(\cdot)|$ in the AIM problem. Afterwards, we treat the set of $|C|$ nodes corresponding to subsets in $C$ as the training set and the rest of $|U|$ nodes either belong to the validation set or the test set. In the AIM problem, we are only allowed to choose training nodes for labeling. Thus, the goal is to select a set $S^*$ of nodes corresponding to subsets in $C$ such that $|\delta_k(S^*)|$ is maximized, which is equivalent to finding the optimal collection $O^*$ in the maximum coverage problem because $K = 1$ and thus $\delta_k(i))$ corresponds to $L_i$. The optimal solution of this instance of the AIM problem implies an optimal solution to the corresponding maximum coverage instance. Since the reduction can be performed in polynomial time, the AIM problem is NP-hard. □

THEOREM 2. *The function* $Q(\cdot)$ *is monotone. That is, for any* $S_1 \subseteq S_2$, $Q(S_1) \leq Q(S_2)$.

THEOREM 3. *The function* $Q(\cdot)$ *is submodular. That is, for any* $S_1 \subseteq S_2$ *and* $u \in V \setminus S_2, Q(S_1 \cup \{u\}) - Q(S_1) \geq Q(S_2 \cup \{u\}) - Q(S_2)$.

## 4.5 The Solution

Given that our objective function is submodular and monotone, based on the proof in [32], we can adopt the greedy algorithm which iteratively selects a node with the maximum marginal gain to produce solutions with a $(1 - 1/e)$ approximation ratio.

The most straightforward solution called NaiveGreedy, whose pseudocode is in Appendix B in the supplementary material, requires to update and compare the marginal gain of every node $v$ not in the solution at the each iteration. This strategy incurs $O(|V|^2)$ time cost and is infeasible in practice.

Inspired by an outbreak detection technique [28], we propose an advanced greedy strategy, GreedywithEarlyTermination (GreedyET), whose pseudocode is in Appendix B in the supplementary material, to speedup NaiveGreedy. Specifically, this advanced strategy notably reduces the number of marginal gain computation in each iteration by leveraging the submodularity of our objective function.

The intuition of GreedyET is that many nodes bring very small marginal gains of the involvement score such that they can be easily pruned at subsequent iterations. More formally, let $S_i$ be the selected seed set after the $i$-th iteration and $Q_\triangle(v|S_i) = Q(S_i \cup \{v\}) - Q(S_i)$ be the marginal gain of $v$ w.r.t. $S_i$. Based on the submodularity of our objective (i.e., for any $S_i \subseteq S_j$ and $v \in V \setminus S_j, Q(S_i \cup \{v\}) - Q(S_i) \geq Q(S_j \cup \{v\}) - Q(S_j)$ ), we know that $Q_\triangle(v|S_i)$ is an upper bound for any $Q_\triangle(v|S_j)$. Therefore, GreedyET first computes $Q_\triangle(v|\emptyset)$ to select $S_1$ and uses it as the upper bound of $Q_\triangle(v|S_1)$ in the next iteration. At each iteration $j$ ($2 \leq j \leq K$), GreedyET processes each node $v \in V \setminus S_{j-1}$ in a non-increasing order of their upper bounds and computes $Q_\triangle(v|S_{j-1})$. Instead of processing all nodes, GreedyET triggers an early termination whenever the maximum upper bound of unprocessed nodes is smaller than the maximum $Q_\triangle(\cdot|S_{j-1})$ of processed nodes. Then, GreedyET updates the upper bounds of each processed node $v$ as $Q_\triangle(v|S_{j-1})$ and proceeds to the next iteration. Although GreedyET does not improve the worst-case time complexity, it is much more efficient than NaiveGreedy in practice (e.g., 94x speedups in our experiments).

## 5 EXPERIMENT

### 5.1 Experimental Setup

**Datasets**. We use six popular datasets for node classification, namely Cora [34], CiteSeer [34], PubMed [34], Coauthor Physics [36] (CoPhy), Flickr [44] and Ogbn-arxiv [22] (Arxiv). For evaluation, we follow the public and commonly used training/validation/test split strategy for these datasets (i.e., following [25] for Cora, CiteSeer, PubMed, [30] for Coauthor Physics, [44] for Flickr and [22] for Ogbn-arxiv). The dataset statistics can be found in Appendix C.1.
**Methods for Comparison**. We compare our methods, NaiveGreedy and GreedyET, against existing methods AGE [6], ANRMAB [16], and GRAIN [48] with their original implementations. Specifically, GRAIN includes two kinds of approaches, namely GRAIN(NN-D) and GRAIN(ball-D), which calculate the diversity

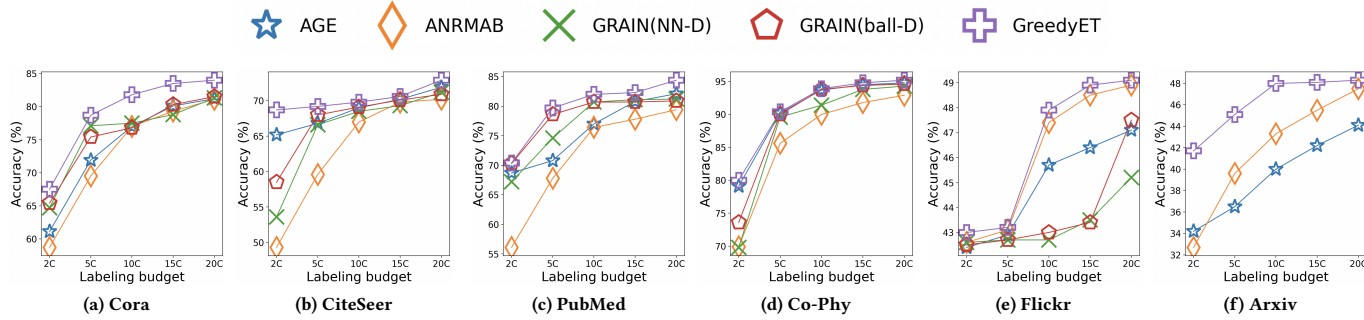

**Figure 4: Test accuracy comparison with various labeling budgets. Grain cannot scale at Arxiv due to out of memory[1].**

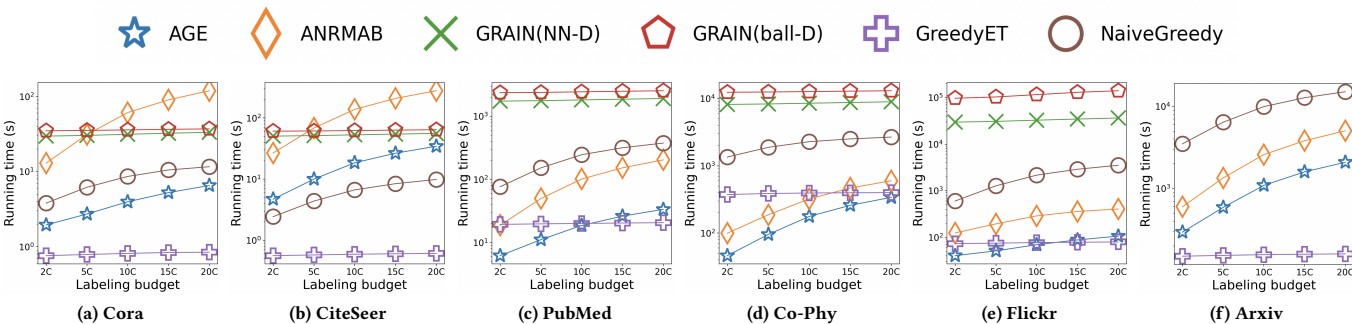

**Figure 5: Running time comparison with different labeling budgets.**

score in different ways. Their description can be found in Related Work and Appendix C.2. Note that, since NaiveGreedy produces the same solution as GreedyET but in a slower way, we only show the performance of NaiveGreedy in the efficiency study (Exp 2).

**GNN Models**. We perform evaluations with four popular GNN models, namely GCN [25], APPNP [26], GraphSAGE (GS) [19] and SGC [39]. Their detailed description can be found in Appendix C.3. Following existing literature [6, 16, 48] in Active Learning, we adopt GCN as the default GNN model and evaluate performance with other models in the flexibility study.

**Parameter Settings**. For our method, we perform grid search of $k$ in $\delta_k(\cdot)$ over the set $\{1, 2\}$ and the similarity threshold $t$ over the set $\{0.1, 0.3, 0.5, 0.7, 0.9, 0.9999\}$, and use cosine similarity to compute the involvement-based similarity. For baseline methods, the recommended setup is adopted. For the labeling budget $B$, we follow previous work [16, 48] by varying the budget $B$ from $2C$ to $20C$ where $C$ is the number of classes in the dataset (e.g., $C = 7$ in Cora) and $20C$ is the default budget. Our code is available at https://gitfront.io/r/user-1291570/gC9qPniGQKs2/AL-Greedy/.

## 5.2 Experimental Results

**Exp 1 - Accuracy Comparison**. Fig. 4 shows the test accuracy (e.g., the percentage of correct prediction) achieved by different methods with different labeling budgets. The results show that our method notably and consistently outperforms baselines. Specifically, GreedyET can achieve up to 7.5% higher accuracy than the second

---

[1] We only consider the case where all necessary data is loaded into memory. Tricks (such as https://github.com/zwt233/Grain/issues/2) that store data in disks, incur high I/O for repetitive data loading and intermediates re-computation are not considered.

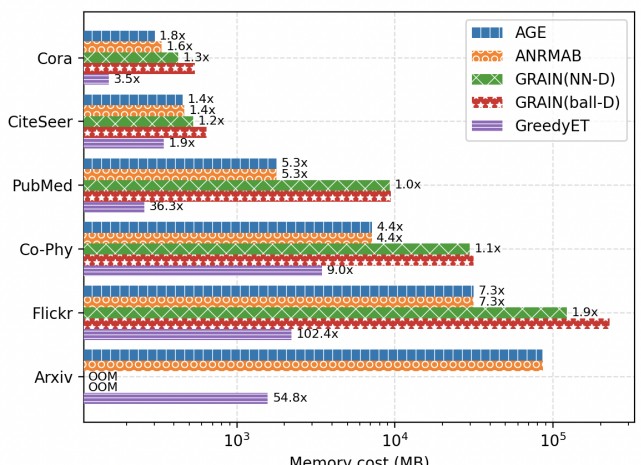

**Figure 6: Memory comparison where the number (e.g., 1x) on each bar shows the advantage (e.g., using 1x less memory) against the worst performer. OOM[1] stands for out of memory.**

best performer (i.e., on Ogbn-arxiv dataset with $2C$ budget) and 19.4% higher accuracy than the worst performer (i.e., on CiteSeer dataset with $2C$ budget).

**Exp 2 - Time Cost Comparison**. Fig. 5 shows the running time of all methods (including preprocessing) with various labeling budgets $B$. We observe that (1) GreedyET is very efficient and achieves up to five-orders-of-magnitude speedups over baselines (i.e., comparing GRAIN(ball-D) on Flickr with $20C$ budget); (2) due to the superiority

**Table 3: Flexibility test with 20$C$ budget (NN-D, ball-D, ET and OOM stand for GRAIN(NN-D), GRAIN(ball-D), GreedyET and out of memory respectively)**

| Dataset | GNN | Seed Selection Method | | | | |
|---|---|---|---|---|---|---|
| | | AGE | MAB | NN-D | ball-D | ET |
| Cora | GCN | 81.2 | 81.1 | 81.3 | 81.5 | **84.0** |
| | APPNP | 75.5 | 74.2 | 66.8 | 67.6 | **79.2** |
| | GS | 76.2 | 73.9 | 77.7 | 74.3 | **81.7** |
| | SGC | 79.5 | 79.3 | 78.0 | 78.1 | **80.5** |
| CiteSeer | GCN | 71.9 | 70.2 | 71.2 | 70.9 | **72.9** |
| | APPNP | 68.5 | 66.0 | 68.7 | 70.1 | **70.7** |
| | GS | 67.7 | 64.3 | 65.3 | 67.2 | **68.0** |
| | SGC | 69.6 | 68.9 | 68.2 | 68.5 | **70.0** |
| PubMed | GCN | 82.1 | 79.4 | 81.2 | 80.9 | **84.4** |
| | APPNP | 80.6 | 73.0 | 81.1 | 80.6 | **83.6** |
| | GS | 77.0 | 75.0 | 76.6 | 74.5 | **80.8** |
| | SGC | 77.6 | 76.5 | 78.7 | 78.8 | **80.9** |
| Co-Phy | GCN | 94.8 | 92.9 | 94.3 | 94.6 | **95.2** |
| | APPNP | 93.5 | 94.0 | 94.4 | 94.8 | **95.1** |
| | GS | 94.0 | 92.0 | 93.0 | 93.5 | **94.4** |
| | SGC | 94.7 | 92.0 | 93.9 | 93.9 | **95.0** |
| Flickr | GCN | 47.1 | 48.9 | 45.2 | 47.5 | **49.1** |
| | APPNP | 42.3 | 44.9 | 42.3 | 42.3 | **46.3** |
| | GS | 42.9 | 42.7 | 42.3 | 42.7 | **44.2** |
| | SGC | 46.8 | 48.8 | 44.7 | 43.5 | **49.2** |
| Arxiv | GCN | 44.1 | 47.5 | OOM | OOM | **48.3** |
| | APPNP | 51.8 | 52.5 | OOM | OOM | **53.6** |
| | GS | 44.3 | 46.7 | OOM | OOM | **48.7** |
| | SGC | 37.9 | 41.1 | OOM | OOM | **41.2** |

**Table 4: Consideration Study (CX means Consideration X)**

| | Cora | CiteSeer | PubMed | Co-Phy | Flickr | Arxiv |
|---|---|---|---|---|---|---|
| C1 | 82.5 | 71.1 | 81.6 | 94.7 | 44.6 | 47.3 |
| C2 | 82.8 | 71.8 | 81.8 | 94.9 | 47.7 | 47.8 |
| C3 | **84.0** | **72.9** | **84.4** | **95.2** | **49.1** | **48.3** |

of the early termination technique, the running time of GreedyET is not sensitive to the labeling budget and is 4x - 94x faster than NaiveGreedy; (3) efficient baselines like AGE and ANRMAB are very sensitive to the labeling budget such that their efficiency will degrade notably as the budget increases.

**Exp 3 - Memory Cost Comparison**. Fig. 6 compares the memory cost of all methods with the labeling budget $B = 20C$. Since baselines require expensive matrix operations or model training for seed selection, their memory footprints are notably larger than our method. Specifically, GreedyET saves up to 102.4x memory space than the worst performer (i.e., on Flickr) and 1.3x than the second best performer (i.e., on CiteSeer), demonstrating its high scalability and feasibility in practice.

**Exp 4 - Flexibility Test**. To evaluate the flexibility of different methods, we evaluate the selected seeds under different GNNs with the default budget, as shown in Table 3. The results show that GreedyET consistently outperforms baselines, which indicates the effectiveness of our objective goal in terms of capturing the commonalities of popular GNNs and strong flexibility of GreedyET.

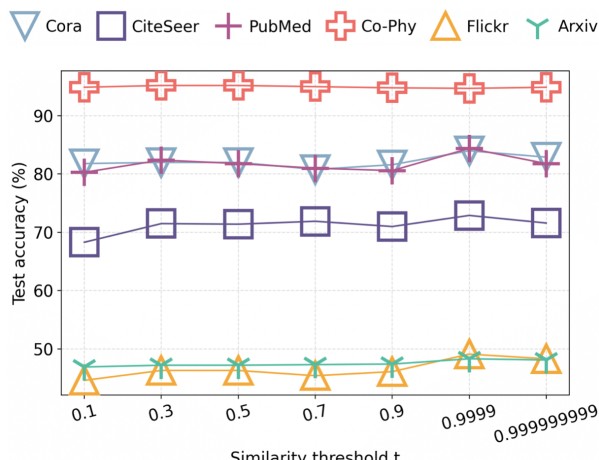

**Figure 7: Hyper-parameter study on $t$**

**Table 5: Hyper-parameter study on $k$**

| Dataset | Accuracy (%) | | Time (s) | |
|---|---|---|---|---|
| | $k = 1$ | $k = 2$ | $k = 1$ | $k = 2$ |
| Cora | 84.0 | 84.1 | 0.8 | 2.7 |
| CiteSeer | 72.9 | 73.5 | 0.6 | 1.5 |
| PubMed | 84.4 | 84.1 | 20.7 | 126.7 |
| Co-Phy | 91.6 | 95.2 | 41.0 | 399.4 |
| Flickr | 49.1 | 49.7 | 79.7 | 3344.0 |
| Arxiv | 48.3 | 48.5 | 159.7 | 6715.3 |

**Exp 5 - Consideration Study**. We validate the effectiveness of different considerations for designing the indirect involvement score in Section 4.3, as shown in Table 4. The result shows that the accuracy gradually increases when we have more considerations and normalization is very important and greatly increases the accuracy.

**Exp 6 - Hyper-parameter Study**. Figure 7 and Table 5 shows the impacts of the similarity threshold $t$ and the number $k$ of hops of the aggregation neighborhood $\delta_k(\cdot)$ respectively. The results show that (1) the performance generally increases and reaches the peak when $t = 0.9999$ and then decreases if $t$ is very close to 1, (2) setting $t = 0.9999$ enables the similarity increment to enjoy a wide range of gradients of the indirect involvement function and can better encourage aggregation neighborhood diversification in a fine-grained manner, and (3) accuracy at $k = 1$ can be very competitive with accuracy at $k = 2$ in many cases but the runtime cost at $k = 1$ can be notably smaller.

## 6 CONCLUSION

In this paper, we tackle the active learning problem by reformulating it as the aggregation involvement maximization problem and proposing an unsupervised, scalable and flexible method with theoretical guarantees. Extensive experiments demonstrate the efficiency, effectiveness, scalability and flexibility of our method. Currently, our method does not support incremental computation for dynamic graphs. In future, we will explore how to partially update the results when the graph structure is changed.

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

## A PROOF OF THEOREM 2 AND THEOREM 3

We prove Theorem 2 and Theorem 3 via proving several lemmas.

LEMMA 1. *The function $|\delta_k(\cdot)|$ is monotonically non-decreasing such that, for any $S_1 \subseteq S_2$ and $v \in V \setminus S_2$, $|\delta_k(S_1)| \le |\delta_k(S_2)|$.*

PROOF. Since $\delta_k(S_1) = \cup_{s \in S_1} \delta_k(s) \subseteq \cup_{s \in S_2} \delta_k(s) = \delta_k(S_2)$, the lemma is deduced. □

LEMMA 2. *The function $|\delta_k(\cdot)|$ is submodular such that, for any $S_1 \subseteq S_2$ and $u \in V \setminus S_2$, $|\delta_k(S_1 \cup \{u\})| - |\delta_k(S_1)| = |\delta_k(S_1 \cup \{u\})) \setminus \delta_k(S_1)| \ge |\delta_k(S_2 \cup \{u\})| - |\delta_k(S_2)| = |\delta_k(S_2 \cup \{u\}) \setminus \delta_k(S_2)|$.*

PROOF. $\delta_k(S_1 \cup \{u\}) \setminus \delta_k(S_1)$ refers to the set of elements that are in $\delta_k(\{u\})$ but are not in the union $\cup_{s \in S_1} \delta_k(s)$. Clearly, it is at least as large as the set of elements that are in $\delta_k(\{u\})$ but are not in the (greater) union $\cup_{s \in S_2} \delta_k(s)$. That is, $\delta_k(S_1 \cup \{u\}) \setminus \delta_k(S_1) \supseteq \delta_k(S_2 \cup \{u\}) \setminus \delta_k(S_2)$. Therefore, the lemma is deduced. □

LEMMA 3. *The function $SIM_v(\cdot)$ is monotonically non-decreasing. That is, for any $S_1 \subseteq S_2$ and $v \in V \setminus S_2$, $SIM_v(S_1) \le SIM_v(S_2)$.*

PROOF. Since $SIM_v(S_1) = \sum_{r \in \delta_k(S_1) \cap \mathcal{H}(v)} sim_v(r)$ and $\delta_k(S_1) \cap \mathcal{H}(v) \subseteq \delta_k(S_2) \cap \mathcal{H}(v)$, the lemma is deduced. □

LEMMA 4. *The function $SIM_v(\cdot)$ is submodular. That is, for any $S_1 \subseteq S_2$ and $u, v \in V \setminus S_2$, $SIM_v(S_1 \cup \{u\}) - SIM_v(S_1) \ge SIM_v(S_2 \cup \{u\}) - SIM_v(S_2)$.*

PROOF. Since

$$SIM_v(S_1 \cup \{u\}) - SIM_v(S_1) = \sum_{r \in \delta_k(S_1 \cup \{u\}) \setminus \delta_k(S_1) \cap \mathcal{H}(v)} sim_v(r),$$

and, based on Lemma 2, we have

$$\delta_k(S_1 \cup \{u\}) \setminus \delta_k(S_1) \cap \mathcal{H}(u) \supseteq \delta_k(S_2 \cup \{u\}) \setminus \delta_k(S_2) \cap \mathcal{H}(u).$$

Thus, the lemma is deduced. □

LEMMA 5. *The function $I_v(SIM_v(\cdot))$ is submodular. That is, for any $S_1 \subseteq S_2$ and $u, v \in V \setminus S_2$, $I_v(SIM_v(S_1 \cup \{u\})) - I_v(SIM_v(S_1)) \ge I_v(SIM_v(S_2 \cup \{u\})) - I_v(SIM_v(S_2))$.*

PROOF. Since $SIM_v(\cdot)$ is monotonically non-decreasing, $I_v(\cdot)$ is also monotonically non-decreasing but its gradient decreases as the input becomes larger. The decreasing property of the gradient indicates that, given the same increment to $SIM_v(S_1)$ and $SIM_v(S_2)$, the increment $I_v(\cdot)$ achieved by the former case will be at least as large as the one achieved by the latter case. Based on Lemma 4, the increment brought by $SIM_v(S_1 \cup \{u\})$ to $SIM_v(S_1)$ will be at least as large as the increment brought by $SIM_v(S_2 \cup \{u\})$ to $SIM_v(S_2)$. Therefore, the lemma is deduced. □

### A.1 Proof of Theorem 2

PROOF. When we change the solution from $S_1$ to $S_2$, all nodes can be divided into $\delta_k(S_1)$, $\delta_k(S_2) \setminus \delta_k(S_1)$ and $V \setminus \delta_k(S_2)$. The contribution of a node $v$ to $Q(\cdot)$ remains as 1 if $v \in \delta_k(S_1)$, becomes 1 if $v \in \delta_k(S_2) \setminus \delta_k(S_1)$, and becomes $I_v(SIM_v(S_2))$ if $v \in V \setminus \delta_k(S_2)$. Thus,

$$Q(S_2) - Q(S_1) = \sum_{v \in \delta_k(S_2) \setminus \delta_k(S_1)} (1 - I_v(SIM_v(S_1)))$$
$$+ \sum_{v \in V \setminus \delta_k(S_2)} (I_v(SIM_v(S_2)) - I_v(SIM_v(S_1))). \quad (7)$$

Clearly, the two terms at the right side of the equation above are both greater or equal to 0. Thus, the theorem is deduced. □

### A.2 Proof of Theorem 3

PROOF. When the solution changes from $\delta_k(S_2)$ to $\delta_k(S_2 \cup \{u\})$, all nodes can be divided into three sets, namely $\delta_k(S_2)$, $\delta_k(S_2 \cup \{u\}) \setminus \delta_k(S_2)$, and $V \setminus \delta_k(S_2 \cup \{u\})$. The contribution of a node $v$ to $Q(\cdot)$ remains as 1 if $v \in \delta_k(S_2)$, becomes 1 if $v \in \delta_k(S_2 \cup \{u\}) \setminus \delta_k(S_2)$ and becomes $I_v(SIM_v(S_2 \cup \{u\}))$ if $v \in V \setminus \delta_k(S_2 \cup \{u\})$. Thus we have

$$Q(S_2 \cup \{u\}) - Q(S_2) = \sum_{v \in \delta_k(S_2 \cup \{u\}) \setminus \delta_k(S_2)} (1 - I_v(SIM_v(S_2)))$$
$$+ \sum_{v \in V \setminus \delta_k(S_2 \cup \{u\})} (I_v(SIM_v(S_2 \cup \{u\})) - I_v(SIM_v(S_2)))$$

Similarly, we have

$$Q(S_1 \cup \{u\}) - Q(S_1) = \sum_{v \in \delta_k(S_1 \cup \{u\}) \setminus \delta_k(S_1)} (1 - I_v(SIM_v(S_1)))$$
$$+ \sum_{v \in V \setminus \delta_k(S_2 \cup \{u\})} (I_v(SIM_v(S_1 \cup \{u\})) - I_v(SIM_v(S_1)))$$
$$+ \sum_{v \in \delta_k(S_2 \cup \{u\}) \setminus \delta_k(S_1 \cup \{u\})} (I_v(SIM_v(S_1 \cup \{u\})) - I_v(SIM_v(S_1)))$$

Since $|\delta_k(S_1 \cup \{u\}) \setminus \delta_k(S_1)| \ge |\delta_k(S_2 \cup \{u\}) \setminus \delta_k(S_2)|$ and $I_v(SIM_v(S_1)) \le I_v(SIM_v(S_2))$, we have $\sum_{v \in \delta_k(S_1 \cup \{u\}) \setminus \delta_k(S_1)} (1 - I_v(SIM_v(S_1))) \ge \sum_{v \in \delta_k(S_2 \cup \{u\}) \setminus \delta_k(S_2)} (1 - I_v(SIM_v(S_2)))$. Since

$$\sum_{v \in V \setminus \delta_k(S_2 \cup \{u\})} (I_v(SIM_v(S_1 \cup \{u\})) - I_v(SIM_v(S_1))) \ge$$
$$\sum_{v \in V \setminus \delta_k(S_2 \cup \{u\})} (I_v(SIM_v(S_2 \cup \{u\})) - I_v(SIM_v(S_2))) \text{ (Lemma 5)},$$
$$\sum_{v \in \delta_k(S_2 \cup \{u\}) \setminus \delta_k(S_1 \cup \{u\})} (I_v(SIM_v(S_1 \cup \{u\})) - I_v(SIM_v(S_1))) \ge 0,$$
$$(8)$$

we have $Q(S_1 \cup \{u\}) - Q(S_1) \ge Q(S_2 \cup \{u\}) - Q(S_2)$. □

## B PSEUDOCODE OF NAIVEGREEDY AND GREEDYET

Algorithm 1 and Algorithm 3 show the pseudocodes of NaiveGreedy and GreedyET respectively.

---

**Algorithm 1:** NaiveGreedy

**Input**  : The input network $G = (V, E)$, the budget $B$ and an integer $k$.

**Output**: The seed set $S$.

1 **foreach** $v \in V$ **do**
2 $\quad$ Conduct BFS to obtain $\delta_k(v)$ and $\mathcal{H}(v)$;
3 $\quad$ Compute $DU_v$ and $NF_v$ based on $\mathcal{H}(v)$;
4 $S \leftarrow \emptyset$;
5 **while** $|S| < B$ **do**
6 $\quad$ $maxgain = 0$;
7 $\quad$ **foreach** $v \in V \setminus S$ **do**
8 $\quad\quad$ $Q_\triangle(v|S) = ComputeGain(G, v, S)$ // the marginal gain of involvement score of $v$ wrt. $S$ ;
9 $\quad\quad$ **if** $Q_\triangle(v|S) > maxgain$ **then**
10 $\quad\quad\quad$ $maxgain = Q_\triangle(v|S)$;
11 $\quad\quad\quad$ $s^* = v$;
12 $\quad$ $S = S \cup \{s^*\}$;
13 Return $S$;

---

**Algorithm 2:** ComputeGain

**Input**  : The input network $G = (V, E)$, the candidate node $v$ and the current seed set $S$.

**Output**: The marginal gain $Q_\triangle$ of $v$ over $S$.

1 $L \leftarrow \emptyset$; // it stores the set of nodes whose indirection involvement scores need to be updated;
2 $\delta_{k\triangle}(v|S) = \delta_k(\{v\}) - \delta_k(S)$ // the marginal gain of direct involvement;
3 $I_\triangle(v|S) = 0$ // the marginal gain of indirect involvement;
4 **foreach** $r \in \delta_{k\triangle}(v|S)$ **do**
5 $\quad$ **foreach** $u \in \mathcal{H}(r) \setminus \delta_k(S \cup \{v\})$ **do**
6 $\quad\quad$ **if** $u \notin L$ **then**
7 $\quad\quad\quad$ $L \leftarrow L \cup \{u\}$;
8 $\quad\quad\quad$ $SIM_u'(S \cup \{v\}) = SIM_u(S)$
9 $\quad\quad$ $SIM_u'(S \cup \{v\}) \mathrel{+}= sim_u(r)$;
10 **foreach** $u \in L$ **do**
11 $\quad$ $I_u(S \cup \{v\}) = \dfrac{2}{1 + e^{-\frac{SIM_u'(S\cup\{v\})}{DU_u} \times NF_u}} - 1$;
12 $\quad$ $I_\triangle(v|S) \mathrel{+}= I_u(S \cup \{v\}) - I_u(S)$;
13 Return $|\delta_{k\triangle}(v|S)| + I_\triangle(v|S)$;

---

## C  EXPERIMENT RELATED DETAILS

### C.1  Dataset Description

**Cora**, **CiteSeer**, and **PubMed**[2] are three public citation network datasets. These three datasets consist of publications that are connected together through citation links, where each publication serves as a node and the citation links represent the edges. The node attributes are binary word vectors, and the class labels indicate the topics to which the publications belong.

**Coauthor Physics**[3] is a co-authorship graph based on the Microsoft Academic Graph from the KDD Cup 2016 challenge. In this dataset, each node represents an author, and an edge connects two authors if they have co-authored a paper. The node features represent the paper keywords for each author's papers, and the class labels indicate the most active fields of study for each author.

---

**Algorithm 3:** GreedywithEarlyTermination (GreedyET)

**Input**  : The input network $G = (V, E)$, the budget $B$ and an integer $k$.

**Output**: The seed set $S$.

1 **foreach** $v \in V$ **do**
2 $\quad$ Conduct BFS to obtain $\delta_k(v)$ and $\mathcal{H}(v)$;
3 $\quad$ Compute $DU_v$ and $NF_v$ based on $\mathcal{H}(v)$;
4 $S \leftarrow \emptyset$;
5 $PQ \leftarrow$ an empty priority queue which sorts nodes based on their upper bounds of marginal gains in non-increasing order;
6 **while** $|S| < B$ **do**
7 $\quad$ **if** $S = \emptyset$ **then**
8 $\quad\quad$ **foreach** $v \in V$ **do**
9 $\quad\quad\quad$ $Q_\triangle(v|S) = ComputeGain(G, v, S)$;
10 $\quad\quad\quad$ Insert $v$ into $PQ$ with $v.ub = Q_\triangle(v|S)$;
11 $\quad\quad$ $s^* \leftarrow PQ.pop()$;
12 $\quad$ **else**
13 $\quad\quad$ $maxgain = 0$;
14 $\quad\quad$ **while** $|PQ| > 0$ **do**
15 $\quad\quad\quad$ $v = PQ.pop()$;
16 $\quad\quad\quad$ $v.ub = ComputeGain(G, v, S)$;
17 $\quad\quad\quad$ **if** $v.ub > maxgain$ **then**
18 $\quad\quad\quad\quad$ $maxgain = v.ub$;
19 $\quad\quad\quad\quad$ $s^* = v$;
20 $\quad\quad\quad$ **if** $|PQ| > 0$ and $maxgain > PQ[0].ub$ **then**
21 $\quad\quad\quad\quad$ break;
22 $\quad\quad$ Update $PQ$ with visited nodes excluding $s^*$;
23 $\quad$ $S = S \cup \{s^*\}$;
24 Return $S$;

---

**Flickr**[4] is a dataset which is built by forming links between images sharing common metadata from Flickr. The image data is collected by the SNAP website from four different sources. In this dataset, each node represents an image, and an edge connects two images if they share some common properties (e.g., same geographic location, same gallery, same tags, etc.). The node features are the bag-of-word representations of the images. The tags of each image are merged into 7 classes and the class labels indicate the classes to which the images belong.

**Ogbn-arxiv**[5] is a directed graph based on the citation network of Computer Science arXiv papers. In this dataset, each node represents an arXiv paper, and an directed edge indicates that one paper cites another. The node features are the averaged embeddings of words in the title and abstract of the arXiv papers, and the class labels indicate the subject areas of the arXiv papers.

The statistics of datasets can be found in Table 6.

### C.2  Details of baselines

- AGE [6]: select nodes based on the node informativeness criteria including uncertainty, information density and graph centrality.
- ANRMAB [16]: improves AGE by adopting a multi-armed bandit framework to dynamically adjust the combination weights of the node informativeness criteria.

---

[2]https://github.com/tkipf/gcn/tree/master/gcn/data
[3]https://github.com/shchur/gnn-benchmark/blob/master/data/npz/ms_academic_phy.npz
[4]https://github.com/GraphSAINT/GraphSAINT#datasets
[5]https://ogb.stanford.edu/docs/nodeprop/#ogbn-arxiv

**Table 6: Datasets statistics where #C stands for the number of classes**

| Dataset | Nodes | Edges | Features | #C | Train/Val/Test |
|---|---|---|---|---|---|
| Cora | 2,708 | 5,429 | 1,433 | 7 | 1.2k/0.5k/1.0k |
| CiteSeer | 3,327 | 4,732 | 3,703 | 6 | 1.8k/0.5k/1.0k |
| PubMed | 19,717 | 44,338 | 500 | 3 | 18.2k/0.5k/1.0k |
| Co-Phy | 34,493 | 247,962 | 8,415 | 5 | 20.7k/6.9k/6.9k |
| Flickr | 89,250 | 899,756 | 500 | 7 | 44.6k/22.3k/22.3k |
| Arxiv | 169,343 | 1,166,243 | 128 | 40 | 90.9k/29.8k/48.6k |

- GRAIN [48]: selects nodes based on the weighted average of the influence score and diversity score. GRAIN includes two kinds of approaches, namely GRAIN(NN-D) and GRAIN(ball-D), which calculate the diversity score in different ways.
- NaiveGreedy: our proposed method which adopts the greedy algorithm to select nodes based on the total direct and indirect involvement scores.

- GreedyET: our improved version of NaiveGreedy with the early termination technique.

## C.3    Details of the compared GNNs
- GCN [25]: extracts features and generate node embeddings by performing a linear approximation to spectral graph convolutions.
- APPNP [26]: leverages the relationship between GCN and PageRank to derive an improved propagation procedure using personalized PageRank with fast approximation.
- GraphSAGE (GS) [19] : constructs node embeddings by sampling and aggregating the feature representations of a node's local neighborhood.
- SGC [39]: the simplified version of GCN which reduces the excess complexity by iteratively eliminating nonlinearities and collapsing weight matrices between consecutive layers.

