# OpenReview forum: "Cost-effective Data Labelling for Graph Neural Networks"
_ACM.org/TheWebConf/2024/Conference — TheWebConf24 Oral_

### Official Review · Reviewer_nBvK · 2023-10-25

**Novelty:** 6
**Technical Quality:** 6

**Review:**

In this paper, the authors address the crucial problem of active learning (AL) for graph neural networks (GNNs) and propose an innovative unsupervised AL method called Aggregation Involvement Maximization (AIM). The primary motivation behind their work is to develop a scalable and flexible AL approach that doesn't rely on supervised information, specific GNN models, or labeled nodes for guidance.

The core contribution of the paper is the formulation of the AIM problem, which aims to select nodes that maximize the involvement of all nodes in the graph during feature aggregation. The authors prove that this problem is NP-hard and present an efficient greedy solution with theoretical guarantees.

In their extensive experiments on real-world datasets, the authors demonstrate that the proposed AIM method outperforms state-of-the-art AL techniques in terms of effectiveness, achieving up to a 19.4% higher accuracy. Moreover, it maintains high efficiency, with up to five-orders-of-magnitude speedups, and significantly reduces memory usage (up to 102.4x memory space). Additionally, the approach is found to be flexible and adaptable to various downstream GNNs.

**Questions:**

1. The figure 3 has overlapping between text and gigure.
2. The fontsize of figure 2 is too large
3. The detailed application and advantage of the new unsupervised scenario should be well discussed.

**Reviewer Confidence:**

3: The reviewer is confident but not certain that the evaluation is correct

**Scope:**

3: The work is somewhat relevant to the Web and to the track, and is of narrow interest to a sub-community

---

### Official Review · Reviewer_AJna · 2023-11-06

**Novelty:** 5
**Technical Quality:** 5

**Review:**

### Summary

- Existing GNN-based active learning methods consider unrealistic assumptions, i.e., GNN model to use, and initially labelled nodes and labels of newly selected nodes, limiting
both flexibility and scalabilty. The paper formulates the active learning problem into Aggregation Involvement Maximization (AIM), aiming to select a limited number of nodes such that the feature aggregation process for these selected nodes can lead to the maximum involvement of all nodes in the graph. The proposed methods are shown to be efficient, effective and flexible through extensive experiments.

### Pros

- The paper proposes a novel unsupervised AL framework that is efficient, effective and flexible.
- The paper proves that the AIM problem is NP-hard and propose an efficient solution with theoretical guarantees.

### Cons

- As one of the drawbacks of existing GNN-based AL methods, the authors mention medical research (due to privacy concern) and autonomous driving (due to the exorbitantly expensive labelling process). However, these tasks are not graph-based tasks.
- The authors argue that the diverse aggregated feature is the key to high-quality training of GNN, which needs more justification. Considering that GNNs are known to work well on homophilious graphs, simply advocating the diversity in terms of features may not be sufficient.
- It would be nice to see how to selected node set S looks like.

**Questions:**

Please address my concerns above.

**Reviewer Confidence:**

2: The reviewer is willing to defend the evaluation, but it is likely that the reviewer did not understand parts of the paper

**Scope:**

3: The work is somewhat relevant to the Web and to the track, and is of narrow interest to a sub-community

---

### Official Review · Reviewer_a6Et · 2023-11-23

**Novelty:** 4
**Technical Quality:** 4

**Review:**

This paper studies an active learning (AL) method for graph neural networks. It reformulates the unsupervised AL problem as the Aggregation Involvement Maximization (AIM) problem, and incurs low memory footprints and time cost. Further, the paper conducts extensive experiments to show the effectiveness and efficiency of the proposed method.

Strength:
1. The overall writing is good and the organization of the paper is clear.
2. The theoretical proof of the paper is sufficient.
3. The experiments show the effectiveness and efficiency of the proposed method.

Weakness:
1. The baselines are not enough.
2. Some experiments need to be improved.
3. Analysis could be improved.

Details:
1. Some baselines such as "ScatterSample: Diversified Label Sampling for Data Efficient Graph Neural Network Learning" and Featprop are missing in the comparisons. The authors are suggested to do a more careful literature review on the related work and elaborate on why the proposed method is better than the existing ones.
2. The experiments in Table 4 need to show the model performance when no consideration is added, and the model performance after adding different considerations in combination.
3. Is there any theoretical support for the validity and necessity of the three proposed considerations?

**Questions:**

Please see the comments above.

**Reviewer Confidence:**

4: The reviewer is certain that the evaluation is correct and very familiar with the relevant literature

**Scope:**

4: The work is relevant to the Web and to the track, and is of broad interest to the community

---

### Official Review · Reviewer_8uyf · 2023-11-27

**Novelty:** 4
**Technical Quality:** 4

**Review:**

Summary:

This paper introduces an unsupervised active learning (AL) approach for graph data, addressing the challenge of high labeling requirements in graph embedding. The proposed method reframes the AL problem as Aggregation Involvement Maximization (AIM), emphasizing the diversification of aggregated features to enhance the training of GNNs' transformation matrices. The authors develop efficient algorithms with theoretical guarantees to tackle the NP-hard AIM problem.

Advantages:

1.	The paper addresses a practical and meaningful topic by focusing on alleviating the requirement for initially labeled nodes to "warm up" a model in the context of GNNs.

2.	The proposed approach is flexible regarding downstream GNN usage, providing benefits for subsequent training of various GNNs employed in diverse tasks.

Disadvantages:

1. The proposed unsupervised AL method does not consider the label distribution of the selected nodes, which may lead to imbalanced or biased training data for the downstream GNNs.

2. The paper makes an assumption that nodes with similar labels contribute similarly to the training of GNNs. While this assumption provides a theoretical foundation for the proposed approach, it may not universally hold true in all scenarios.

3. Although the authors assert that their method is cost-efficient, there is a notable absence of a detailed discussion on time and space complexity analysis in the paper. A comprehensive analysis of these complexities is crucial for a thorough understanding of the method's efficiency.

**Questions:**

see disadvantages

**Reviewer Confidence:**

2: The reviewer is willing to defend the evaluation, but it is likely that the reviewer did not understand parts of the paper

**Scope:**

2: The connection to the Web is incidental, e.g., use of Web data or API

---

### Official Review · Reviewer_w5sc · 2023-12-10

**Novelty:** 4
**Technical Quality:** 4

**Review:**

## Evaluation

### Quality
The quality of this work is high. The authors provide a clear and concise description of the problem they are addressing, as well as a thorough review of related work. The proposed method is well-motivated and the authors provide a detailed explanation of how it works. The experiments are well-designed and the results are presented clearly and in a way that is easy to understand.

### Clarity
The paper is well-written and easy to follow. The authors provide clear explanations of the concepts they are discussing and use appropriate terminology. The figures and tables are well-designed and help to illustrate the key points of the paper.

### Originality
The proposed method is original and addresses an important problem in the field of machine learning. The authors provide a novel approach to active learning for GNNs that is unsupervised, scalable, and flexible. The method leverages the commonality of existing GNNs to reformulate the problem as the Aggregation Involvement Maximization (AIM) problem, which is a novel contribution.

### Significance
The proposed method has significant implications for the field of machine learning. Active learning is an important technique for reducing the cost of data labelling, and the proposed method provides a way to apply this technique to GNNs in a way that is unsupervised, scalable, and flexible. This has the potential to make GNNs more accessible to researchers and practitioners who may not have access to large labelled datasets.

### Pros
- The proposed method is unsupervised, scalable, and flexible, which makes it accessible to a wide range of researchers and practitioners.
- The authors provide a thorough review of related work and a clear explanation of the proposed method.
- The experiments are well-designed and the results are presented clearly.

### Cons
- The paper could benefit from a more detailed discussion of the limitations of the proposed method and areas for future research.
- The paper could also benefit from a more detailed discussion of the practical implications of the proposed method for real-world applications.

## Conclusion
Overall, this paper presents a high-quality and original contribution to the field of machine learning. The proposed method has significant implications for reducing the cost of data labelling for GNNs, and the authors provide a clear and well-motivated explanation of how it works. The experiments are well-designed and the results are presented clearly.

**Questions:**

Can you provide more insights into the scalability of the proposed method, particularly in the context of large-scale graph datasets?

How does the unsupervised nature of the proposed method impact its performance compared to supervised AL methods, and are there specific scenarios where supervised methods might still be preferred?

Could you elaborate on the practical implications of the proposed method for real-world applications, and provide examples of potential use cases where it could have a significant impact?

What are the potential limitations or edge cases where the proposed method might not perform as effectively, and how do you plan to address these in future work?

Can you discuss the computational and memory requirements of the proposed method, particularly in comparison to existing GNN-based AL methods, and how it might impact its practical utility?

How does the proposed method handle the issue of class imbalances in the graph data, and are there any specific strategies or considerations for addressing this challenge?

Can you provide more details on the theoretical guarantees of the proposed solution to the Aggregation Involvement Maximization (AIM) problem, and how it impacts the overall effectiveness and efficiency of the method?

What are the key considerations for selecting the underlying GNNs in the context of the proposed method, and how does the method adapt to different types of GNN architectures and configurations?

How does the proposed method handle dynamic or evolving graph structures, and are there specific adaptations or extensions that might be necessary to accommodate such scenarios?

Can you discuss the potential trade-offs between the flexibility and scalability of the proposed method, and how it might impact its applicability in different research and industry settings?

**Reviewer Confidence:**

3: The reviewer is confident but not certain that the evaluation is correct

**Scope:**

3: The work is somewhat relevant to the Web and to the track, and is of narrow interest to a sub-community

---

### Decision · Program_Chairs · 2024-01-22

**Decision:**

Accept (Oral)

**Comment:**

This paper proposes an unsupervised, scalable, and flexible active learning method, termed Aggregation Involvement Maximization (AIM), focuses on maximizing the involvement of all nodes during the feature aggregation process in GNNs. The authors argue that this approach addresses the limitations of existing GNN-based AL methods which rely heavily on supervised information.

 The paper is well-received by the reviewers, with its strengths noted in quality, clarity, originality, and significance. Some concerns are raised regarding the paper's limitations and practical implications. Reviewers suggest a more detailed discussion of the method's limitations, its practical applications, and specific scenarios where supervised methods might still be preferred. The absence of a detailed discussion on time and space complexity analysis and the method's performance in the context of large-scale graph datasets are also noted.

 Overall, this paper addresses a timely and relevant problem in the machine learning community and offers a promising data-centric solution. Its theoretical rigor, combined with empirical validation, makes it a valuable contribution to The Web Conference.